# Updated Global Fuel Exploitation Inventory (GFEI) for methane emissions from the oil, gas, and coal sectors: evaluation with inversions of atmospheric methane observations

Tia R. Scarpelli[1,2,*], Daniel J. Jacob[1], Shayna Grossman[2], Xiao Lu[1,3], Zhen Qu[1], Melissa P. Sulprizio[1], Yuzhong Zhang[4,5], Frances Reuland[6], Deborah Gordon[7], John R. Worden[8]

[1]School of Engineering and Applied Science, Harvard University, Cambridge, MA, USA
[2]Department of Earth and Planetary Sciences, Harvard University, Cambridge, MA, USA
[3]School of Atmospheric Science, Sun Yat-sen University, Zhuhai, Guangdong, China
[4]Key Laboratory of Coastal Environment and Resources of Zhejiang Province, School of Engineering, Westlake University, Hangzhou, Zhejiang, China
[5]Institute of Advanced Technology, Westlake Institute for Advanced Study, Hangzhou, Zhejiang, China
[6]Rocky Mountain Institute, Boulder, CO, USA
[7]Rocky Mountain Institute, Providence, RI, USA
[8]NASA Jet Propulsion Laboratory, Pasadena, CA, USA
*Now at University of Edinburgh, Edinburgh, Scotland

*Correspondence to: Tia R. Scarpelli (tia.scarpelli@ed.ac.uk)*

**Abstract.** We present an updated version of the Global Fuel Exploitation Inventory (GFEI) for methane emissions and evaluate it with results from global inversions of atmospheric methane observations from satellite (GOSAT) and in situ platforms (GLOBALVIEWplus). GFEI allocates methane emissions from oil, gas, and coal sectors and subsectors to a 0.1° x 0.1° grid by using the national emissions reported by individual countries to the United

Nations Framework Convention on Climate Change (UNFCCC) and mapping them to infrastructure locations. Our updated GFEI v2 gives annual emissions for 2010-2019 that incorporate the most recent UNFCCC national reports, new oil/gas well locations, and improved spatial distribution of emissions for Canada, Mexico, and China. Russia's oil/gas emissions in its latest UNFCCC report (4.1 Tg a[-1] for 2019) decrease by 83% compared to its previous report while Nigeria's latest reported oil/gas emissions (3.1 Tg a[-1] for 2016) increase sevenfold compared to its previous

report, reflecting changes in assumed emission factors. Global gas emissions in GFEI v2 show little net change from 2010 to 2019 while oil emissions decrease and coal emissions slightly increase. Global emissions from the oil, gas, and coal sectors in GFEI v2 (26, 22, and 33 Tg a[-1], respectively in 2019) are lower than the EDGAR v6 inventory (32, 44, and 37 Tg a[-1] in 2018) and lower than the IEA inventory for oil and gas (38 and 43 Tg a[-1] in 2019) though there is considerable variability between inventories for individual countries. GFEI v2 estimates higher emissions by

country than the Climate TRACE inventory with notable exceptions in Russia, the US, and the Middle East where TRACE is up to an order of magnitude higher than GFEI v2. Inversion results using GFEI as a prior estimate confirm the lower Russian emissions in the latest UNFCCC report but find that Nigeria's reported UNFCCC emissions are too high. Oil/gas emissions are generally underestimated by the national inventories for the highest emitting countries including the US, Venezuela, Uzbekistan, Canada, and Turkmenistan. Offshore emissions tend to

be overestimated. Our updated GFEI v2 provides a platform for future evaluation of national emission inventories reported to the UNFCCC using the newer generation of satellite instruments such as TROPOMI with improved coverage and spatial resolution. This increased observational data density will be especially beneficial in regions

where current inversion systems have limited sensitivity including Russia. Our work responds to recent aspirations of the Intergovernmental Panel on Climate Change (IPCC) to integrate top-down and bottom-up information into the construction of national emission inventories.

## 1 Introduction

Countries under the Paris Agreement must set goals for mitigating greenhouse gas emissions through nationally determined contributions (NDCs). The NDCs often include mitigation targets for methane based on national inventories of current methane emissions from different sectors (COP 2021; COP 2016). These national methane emission inventories are submitted to the United Nations Framework Convention on Climate Change (UNFCCC) and form the framework for methane climate policy. But the inventories rely on simple methods of emissions estimation, in part due to a lack of available data, leading to large uncertainties. Uncertainties are particularly large for the oil/gas sector due to the large number of point sources that have potential for large emissions. These uncertainties are particularly relevant to the Global Methane Pledge by which 110 countries have committed to a 30% reduction in methane emissions from 2020 levels by 2030 (European Commission, 2021). The reduction strategy is difficult to define and success will be difficult to measure if the 2020 emission baseline is uncertain.

Inverse analyses of atmospheric methane observations offer an independent check on the emission inventories (Bergamaschi et al., 2009) but require spatially resolved inventory information that is generally not available from UNFCCC reports. Here we provide this information in a global gridded (0.1° x 0.1°) representation of the UNFCCC-reported national emission inventories for fuel exploitation (oil, gas, and coal) emissions in 2010-2019, updating our previous work for 2016 (Scarpelli et al., 2020a). We compare these national inventories to recent inversions of satellite (GOSAT) and in situ (GLOBALVIEWplus) atmospheric methane observations, and draw implications for improving the inventories.

Oil/gas activities are currently estimated to account for 22% (84 Tg a$^{-1}$, range 72-97 Tg a$^{-1}$) of global anthropogenic methane emissions in 2017 according to emission inventories compiled by the Global Carbon Project (Saunois et al., 2020). The potential for economical mitigation makes the oil/gas sector an attractive target for emission reductions (Alvarez et al., 2018). Individual countries report oil/gas methane emissions to the UNFCCC as part of their national inventories using 'bottom-up' methods that apply emission factors (e.g., mass of methane emitted per unit volume of oil produced) to source activity data (e.g., volume of oil produced per year). Annex I countries must report emissions every year by oil/gas subsector (e.g., oil production). Non-Annex I countries are not required to report emissions every year or by subsector, and many use default emission factors from the Intergovernmental Panel on Climate Change (IPCC, 2006, 2019). Emission factors may vary considerably and the corresponding uncertainties carry over to the national inventory (Scarpelli et al., 2020a).

'Top-down' information from observations of atmospheric methane can help to evaluate and improve the bottom-up national inventories (IPCC, 2019). This is generally done by inverse modeling where an atmospheric transport

model is used to relate methane emissions to atmospheric concentrations (Houweling et al., 2017). The top-down

information on emissions comes from observed atmospheric concentration gradients, hence the need for prior information from a spatially resolved inventory. An optimal estimate of emissions can be determined by error-weighted Bayesian inference combining the information from atmospheric observations with that from the bottom-up inventory (Brasseur and Jacob, 2017). Satellite observations are of particular interest for inverse modeling because of their global continuous coverage (Palmer et al., 2021). They use backscattered solar radiation in the

shortwave infrared to retrieve an atmospheric methane column concentration with near-unit sensitivity down to the surface (Jacob et al., 2016).

National inventories submitted to the UNFCCC do not in general provide the spatial resolution needed for the exploitation of top-down information. An exception is the United Kingdom (UK) which provides a finely gridded

yearly inventory (Defra and BEIS, 2021). A number of studies have spatially allocated national inventories for specific years to enable inversions of atmospheric data including for Australia (Wang and Bentley, 2002), Switzerland (Hiller et al., 2014), the US (Maasakkers et al., 2016), Mexico (Scarpelli et al., 2020b), and Canada (Scarpelli et al., 2022). Scarpelli et al. (2020a) constructed the Global Fuel Exploitation Inventory (GFEI) for 2016 that spatially allocates national oil, gas, and coal methane emissions reported to the UNFCCC to a 0.1° x 0.1° grid,

and supplements information for non-reporting countries. This inventory has been used as prior estimate in a number of inversions (Zhang et al., 2021; Shen et al., 2021; Lu et al., 2021; Qu et al., 2021; Western et al., 2021).

Here we update GFEI to 2019 (Scarpelli et al., 2021) using more recent national emissions submitted to the UNFCCC (2021), describe the 2010-2019 national emission trends based on the UNFCCC reports, and interpret the

results from global inversions of atmospheric methane observations using GFEI as prior estimate. We use the bottom-up information embedded in GFEI, including infrastructure locations, to identify the processes that drive discrepancies between the bottom-up and inversion estimates. Our work provides a step towards the aspiration of IPCC (2019) to integrate top-down and bottom-up information in the construction of national inventories for climate policy.

**2 Updated Global Fuel Exploitation Inventory (GFEI v2)**

**2.1 GFEI v1**

Scarpelli et al. (2020a) constructed the Global Fuel Exploitation Inventory version 1 (GFEI v1) at 0.1° x 0.1° grid resolution by disaggregating the national UNFCCC methane emission reports to oil/gas/coal emission subsectors and then allocating subsector emissions to the appropriate infrastructure locations within each country. GFEI v1 was

constructed for 2016 and includes separate gridded emission data for each oil/gas subsector and emission process (leakage, venting, flaring). In North America GFEI v1 uses the reported UNFCCC emissions, but these national emissions are distributed within each country using the gridded inventories from Sheng et al. (2017) for oil/gas in Canada and Mexico, and Maasakkers et al. (2016) for oil/gas/coal in the US.

Gridded uncertainties were constructed for GFEI v1 by applying subsector specific national scale uncertainties to
       gridded emissions, designating between Annex I and non-Annex I countries. These national scale uncertainties and
       the IPCC emission factor uncertainties used to derive them are shown in Table 1 of Scarpelli et al. (2020a). The
       relative error standard deviations for upstream oil/gas (excluding flaring) are 38-50% for Annex I countries and 38-
       100% for non-Annex I countries.

**2.2 Construction of GFEI v2**

       Here we update GFEI to provide annual gridded oil/gas/coal emissions by subsector for 2010-2019 using the most
       recent national reports to the UNFCCC (2021) as of September 2021 combined with new infrastructure information.
       We refer to this updated inventory as version 2 (v2; Scarpelli et al., 2021).

Following the methods of Scarpelli et al. (2020a), we use 2010-2019 emissions as reported to the UNFCCC for
       Annex I countries as these are available by year and subsector. Countries that report to the UNFCCC as non-Annex I
       countries are only required to report total emissions by sector and do not report every year, so we partition non-
       Annex I emissions to the desired subsector and year. We create our own emission estimates for each year by
       applying IPCC emission factors (IPCC, 2006) to yearly activity data from the US Energy and Information
Administration (EIA, 2021), and we use the relative subsector contributions and trends to disaggregate and update
       the UNFCCC reported emissions. Similar to GFEI v1, we incorporate more detailed emission estimates, when
       available, from the most recent National Communications and Biennial Update Reports of the top-emitting (above 1
       Tg a$^{-1}$) non-Annex I countries. This includes Nigeria, for which emissions were below 1 Tg a$^{-1}$ in GFEI v1 but are
       above 1 Tg a$^{-1}$ in its most recent National Communication (see Section 2.3; Federal Republic of Nigeria, 2020). For
those countries that do not report to the UNFCCC (non-reporting), we estimate emissions using IPCC (2006)
       methods and EIA (2021) activity data. For GFEI v2, we do not use UNFCCC national reports if dated prior to 2000
       and instead use IPCC methods as employed for non-reporting countries. This differs from GFEI v1 and most notably
       affects Iraq, leading to a large difference in Iraq's GFEI v2 emissions compared to GFEI v1 (discussed in Section
       2.3).
       For GFEI v2 we start from the same spatial oil/gas infrastructure information as Scarpelli et al. (2020a) which
       includes oil/gas well locations from Enverus and midstream infrastructure (e.g., processing plants, compressor
       stations, refineries) locations from the Global Oil & Gas Infrastructure (GOGI) inventory and geodatabase (Rose et
       al., 2018; Sabbatino et al., 2017). We update well locations using the more recent data from Enverus (2019) and
continue to use the well locations from Rose (2017) for countries missing from the Enverus database as described by
       Scarpelli et al. (2020a). For all oil/gas infrastructure within each country, we allocate national emissions using the
       density of infrastructure per grid cell (e.g., grid cells with a greater number of wells have higher emissions). We
       allocate downstream (distribution) gas emissions using an updated population density map for 2015 (CIESIN, 2017).
       For coal, we allocate national emissions within each country using the 2018 gridded emissions from EDGAR

version 6 (Crippa et al., 2021; European Commission, 2021) with the exception of the UK where we use EDGAR v4.3.2 (Janssens-Maenhout et al., 2019; European Commission, 2017) as there are no UK coal sources in more recent versions of EDGAR.

For North America and China, we use national scale inventories to distribute the UNFCCC reported emissions within each country as these national inventories include more detailed spatial information than our global datasets. We use the same spatial information as GFEI v1 for the US, including Maasakkers et al. (2016) with additional information for Alaska (Scarpelli et al., 2020a). We use improved bottom-up information for the distribution of oil/gas/coal emissions in Mexico (Scarpelli et al., 2020b) and Canada (Scarpelli et al., 2022), and for the distribution of coal emissions in China (Sheng et al., 2019).


Annex I countries report 'other' oil/gas emissions which Scarpelli et al. (2020a) allocated 50% to wells and 50% to pipelines. For GFEI v2, we distribute 'other' emissions to oil/gas subsectors and their corresponding infrastructure relative to the contribution of each subsector to total oil/gas emissions. The US and Canada are exceptions where we instead attribute all 'other' oil/gas emissions to oil/gas production based on national inventories (EPA, 2020;
Scarpelli et al., 2022).

**2.3 GFEI v2 methane emissions**

Figure 1 shows GFEI v2 methane emissions at 0.1° x 0.1° grid resolution for 2019, totaling 26 Tg a$^{-1}$ for oil, 22 Tg a$^{-1}$ for gas, and 33 Tg a$^{-1}$ for coal. Global emissions by sector and oil/gas subsector are compiled in Table 1. Figure 2 shows emissions for the top emitting countries with China, the US, and Russia together accounting for 39% of
global gas emissions and 79% of global coal emissions while oil emissions more evenly distributed among the top emitting countries. GFEI v2 oil and gas production emissions are 32% and 15% lower, respectively, than in GFEI v1 (Table 1), mainly because of downward revision of Russia's national emissions in its latest UNFCCC (2021) report. Global coal emissions do not change significantly between v1 and v2 for the same year.

Figure S1 shows a comparison of emissions in GFEI v2 and GFEI v1 for 2016, aggregated to 2° x 2.5° grid resolution for visibility. Differences reflect changes to national emissions based on UNFCCC reporting, as well as changes to the distribution of emissions within the countries. The use of the Sheng et al. (2019) inventory for the distribution of China's coal emissions leads to higher emissions in the south and lower in the north, in part due to the inclusion of provincial emission factors. The main countries that revised their UNFCCC emissions between GFEI v1
and v2 are the US, Uzbekistan, Nigeria, and Russia. GFEI v2 oil/gas emissions in the US (7.8 Tg a$^{-1}$ for 2016) are 7% lower than GFEI v1, mainly because of downward revision for the gas production subsector. This downward revision reflects the incorporation of facility-reported oil/gas emissions from the US Greenhouse Gas Reporting Program (GHGRP) and the use of new emission factors based on US GHGRP data and field measurements (EPA, 2020; EPA, 2021). Iraq's emissions have also increased in GFEI v2 due to the use of IPCC Tier 1 methods to
estimate its emissions rather than the pre-2000 UNFCCC reported emissions as used in GFEI v1.

Figure 3 shows national emission factors for oil/gas production as implied by GFEI v2 in combination with EIA oil/gas production statistics (e.g., national oil production emissions in GFEI divided by volume of oil produced). These emission factors vary by five orders of magnitude between countries. Also shown is the range of emission factors provided by IPCC (2006) guidelines, ranging from the lowest value for Developed Countries to the highest value for Developing Countries and Countries with Economies in Transition. The IPCC (2006) emission factors vary by over two orders of magnitude and most countries fit within that range. The IPCC (2006) emission factors for gas production equate to leakage rates of 0.06 to 3.8%, assuming 92% methane gas by volume.

The low emission factors shown in Fig. 3 for some Middle East countries could reflect modern infrastructure, high rates of production per well, and widespread associated gas capture and high efficiency flaring. The dominance of offshore production in countries like Norway and Qatar may also contribute to low emission factors. Iraq's higher emissions in GFEI v2 lead to an oil emission factor similar to its neighbor Iran. The order of magnitude decrease in Russian oil emissions and increase in Nigerian oil emissions between GFEI v1 and v2 (Table 2) reflect a switch in the emission factors used by the national inventories. Nigeria uses an emission factor at the upper limit of the IPCC (2006) range in its most recent report to the UNFCCC (Federal Republic of Nigeria, 2020).

Russia previously used the IPCC (2006) emission factors for Developing Countries and Countries with Economies in Transition (Russian Federation, 2018), but in its most recent report (Russian Federation, 2021) it uses the IPCC (2006) emission factors for Developed Countries and country-specific emission factors based on measurements (mostly limited to gas activities). The methodology update is, in part, based on increases in gas use for energy and rules limiting associated gas flaring (Russian Federation, 2021). Previous inverse studies found that oil/gas emissions in the older Russian national inventory were too high (Maasakkers et al., 2019; Zhang et al., 2021), supporting the decrease in the revised inventory.

Figure 4 shows GFEI v2 emission trends over 2010-2019. These trends are determined using emissions as reported to the UNFCCC (2021) for Annex I countries (in Fig. 4 this includes Russia, the US, and Ukraine) and otherwise using EIA activity data to scale annually the reported inventory years. Global oil emissions show a decrease over 2010-2013 driven by Libya and Iran and over 2017-2019 driven by Venezuela. This global emissions decrease is in contrast to a 14% increase in global oil production (EIA, 2020), and reflects compensation between decreased production in countries with high emission factors like Venezuela and Iran, and increased production in countries with low emission factors like Brazil and Kuwait (Fig. 3).

Global gas emissions decrease over 2011-2017, mostly driven by Russia, and then increase in 2018 and 2019 due to contributions from various countries, including Uzbekistan, the US, and Ukraine. Global coal emissions slightly increase from 2010 to 2019 with large interannual variability mainly driven by China (based on EIA activity data). Coal emissions show a steady decrease in the US and an increase in Russia.

Figure 5 shows global oil, gas, and coal emissions for GFEI along with the most recent estimates from the EDGAR

v6 inventory (Crippa et al., 2021; European Commission, 2021) and from the International Energy Agency (IEA) inventory (IEA, 2021). The IEA inventory does not include coal emissions. GFEI v1 has higher oil emissions than the other bottom-up inventories, mostly attributable to the high Russian emissions mentioned previously. Global emissions in EDGAR and IEA are higher than GFEI v2 for all sectors but with considerable variability between countries including in the sign of the difference as shown in Fig. S2. Iraq's higher emissions in GFEI v2 compared

to GFEI v1 are in better agreement with the other bottom-up inventories though EDGAR and IEA still estimate higher emissions.

## 3 Information from inverse analyses

Here we examine results from two recent global inversions of atmospheric methane observations that used GFEI v1 as a prior estimate of emissions (Lu et al., 2021; Qu et al., 2021), to determine what insights can be gained from

atmospheric methane observations toward improving the bottom-up inventories. We focus our discussion on oil/gas emissions because of the difficulty for these inversions to quantify coal emissions in China (Qu et al., 2021). This difficulty is due in part to poor spatial allocation of Chinese emissions since corrected in GFEI v2 (Sheng et al., 2019).

### 3.1 Methods

Lu et al. (2021; referred to hereafter as Lu21) and Qu et al. (2021; referred to hereafter as Qu21) used similar inversion procedures but applied them to different observations, time periods, and spatial resolution. They also used different prior estimates for wetlands. Both inversions used GEOS-Chem as the forward chemical transport model. Both optimized a state vector $x$ including annual non-wetland emissions on the GEOS-Chem grid, monthly wetland emissions for 14 subcontinental regions (Bloom et al., 2017), and the mean concentration of tropospheric OH (the

main methane sink) in each hemisphere. Lu21 optimized mean non-wetland methane emissions for 2010-2017 and their linear temporal trends on a 4° x 5° grid while Qu21 optimized non-wetland methane emissions for 2019 on a 2° x 2.5° grid

Lu21 used 2010-2017 GOSAT satellite observations of methane columns (Parker et al., 2020) and an ensemble of in

situ measurements of the atmospheric methane concentration from surface sites, aircraft, and ships compiled as the GLOBALVIEWplus $CH_4$ ObsPack v1.0 database (NOAA ESRL, 2019). Qu21 used 2019 GOSAT and TROPOMI satellite observations of methane columns separately and together. The TROPOMI observations in Qu21 were from the early-generation retrieval of Hu et al. (2018) and showed some major regional biases that propagated to the inversion results. Here we focus on their GOSAT-only inversion results. Lu21 excluded GOSAT observations over

the oceans (glint) and both inversions excluded observations poleward of 60°.

For both inversions, gridded non-wetland emissions were assumed to have a prior error standard deviation of 50%. For wetland emissions, Lu21 and Qu21 used prior error variances and covariances from Bloom et al. (2017), but

Qu21 found that they needed to greatly decrease these errors (by a factor of 24) to regularize their inversion of TROPOMI data and they applied the same low prior errors for wetlands in their inversion of GOSAT data.

Both Lu21 and Qu21 used the same analytical solution to minimization of the Bayesian cost function in order to produce their posterior emission estimates (Jacob et al., 2016). The analytical solution provides not only a maximum-probability posterior estimate $\hat{x}$ for the state vector, but also a closed-form posterior error covariance matrix ($\hat{S}$) for that state vector from which we can determine the information content of the inversion using the averaging kernel matrix ($A = I - \hat{S}S_A^{-1}$, where $S_A$ is the prior error covariance matrix). The diagonal terms of $A$ represent the averaging kernel sensitivities ($a_j$) that characterize the ability of the atmospheric observations to determine emissions from grid cell $j$ independently of the prior estimate (perfectly if $a_j = 1$, not at all if $a_j = 0$). The trace of $A$ defines the degrees of freedom for signal (DOFS), representing the number of independent pieces of information on methane emissions that can be obtained from the observations (Rodgers, 2000).

The emissions from a particular sector or subsector can be inferred from the inversion results by applying a summation matrix ($W$):

$$\hat{x}' = W\hat{x} \quad (1),$$
$$A' = WAW^* \quad (2)$$

where $W^* = W^T(WW^T)^{-1}$ is the pseudo inverse matrix (Calisesi et al., 2005). Here $\hat{x}'$ is a posterior state vector of sectoral/subsectoral emissions per grid cell, country, or globally; and $A'$ is the corresponding averaging kernel matrix. $W$ is constructed by using the prior estimates of the sector/subsector fractional contributions to emissions in individual grid cells, and summing those nationally or globally. We use GFEI at the native 0.1° x 0.1° resolution to better resolve boundaries in estimates of national emissions, but the coarse resolution of the inversions is still a limitation for small countries and for oil/gas emissions near country borders. More advanced methods for inferring sectoral emissions from gridded inversion results include consideration of the different prior error estimates for individual sectors (Cusworth et al., 2021; Shen et al., 2021; Worden et al., 2021), but information on these prior error estimates is limited.

Figure S3 shows posterior oil/gas emissions and averaging kernel sensitivities for the Lu21 and Qu21 inversions. Lu21 report a global DOFS of 262 for optimizing non-wetland emissions on their 4° x 5° grid while Qu21 report a DOFS of 232 on their 2° x 2.5° grid. The higher resolution and low wetland prior errors in Qu21 would be expected to lead to higher DOFS, but this is offset by the use of 8 years of both satellite and in situ data in Lu21, with the inclusion of the in situ data increasing DOFS by 25% compared to the GOSAT-only result.

### 3.2 Results and discussion

Figure 5 shows global oil, gas, and coal emissions from the inversions and Table 1 gives further detail for oil/gas
subsectors. Lu21 emissions are their mean values for 2010-2017. Global gas emissions in Lu21 and Qu21 are 23-
36% higher than GFEI v2 with higher emissions for all gas subsectors (Table 1). Averaging kernel sensitivities are
high for upstream gas activities (production and processing) but low for gas transmission and distribution. Lu21 and
Qu21 estimate much lower gas emissions compared to EDGAR and IEA estimates (Fig. 5), and averaging kernel
sensitivities are sufficiently high that this difference cannot be simply attributed to the lower prior estimate. Global
oil emissions in Lu21 are slightly lower than GFEI v1 (7% lower) while Qu21 emissions are much lower (34%) and
in better agreement with GFEI v2, mostly due to decreases in Russian oil emissions. Global oil emissions in
EDGAR and IEA are in between the Lu21 and Qu21 estimates. Although trends in global oil and gas emissions may
contribute to differences between the Lu21 results for 2010-2017 and the GFEI v2 and Qu21 results for 2019 (Fig.
4), GFEI v2 trends imply that this impact is likely small (on the order of 2 Tg; Fig. 4).

Figure 6 compares the national oil/gas emissions in the Lu21 and Qu21 inversions to the different bottom-up
inventories. We also compare to oil/gas methane emission estimates in the Climate TRACE inventory (Reuland et
al., 2021). The TRACE inventory provides annual country-level emission estimates for oil and gas production,
processing and distribution, and oil refining which are generated with the Oil Climate Index + Gas (OCI+), an open-
source, bottom-up systems tool (Gordon et al., 2015). Both inversions use GFEI v1 as a prior estimate, so results are
directly relevant to evaluating the national reports to the UNFCCC. The averaging kernel sensitivities in Fig. 6
indicate the dependence of the inversion results on the prior estimate (1 = totally independent, 0 = totally
dependent). They are generally related to the density of observations, which for GOSAT is mainly limited by cloud
cover and high latitudes (>60°), with higher density for the US and Canada in Lu21 because of the
GLOBALVIEWplus surface sites. Even when averaging kernel sensitivities are low the sign of the corrections
relative to GFEI v1 is informative. There are some large discrepancies between Lu21 and Qu21, generally for
countries with low averaging kernel sensitivities in Qu21. An additional concern with Qu21 is the strong prior
constraint on wetland emissions that may lead to aliasing of wetland emissions adjustments to oil/gas when there is
spatial overlap (such as Russia and Canada). We therefore focus on the Lu21 results but add the perspective from
the Qu21 results when appropriate.

Table 2 shows Lu21 oil/gas emissions by country for the top-emitting countries which account for 82% of Lu21
global oil/gas emissions. Upstream oil/gas activities (oil/gas production and gas processing; Table S1 and S2) have
the largest emissions contribution. The inversion finds emission underestimates in GFEI v1 and v2 for these top-
emitting countries including the US, Venezuela, Uzbekistan, Canada, and Turkmenistan, with Russia as the major
exception. Correcting emissions in these countries leads to the higher global gas emissions in Lu21 compared to
GFEI v1 and v2. The global oil emissions in Lu21 show little change from GFEI v1 because Venezuela's emissions
increase is offset by the large decrease in Russian oil emissions.

Russia accounts for 25% of global oil/gas emissions in the Lu21 inversion, with a national total of 15.8 Tg a$^{-1}$. This is lower than GFEI v1 (24.9 Tg a$^{-1}$), used as prior estimate, but still higher than the other bottom-up inventories including GFEI v2 for 2016 (4.3 Tg a$^{-1}$). Averaging kernel sensitivities for Russia are relatively low in the Lu21 inversion because the high latitude oil/gas emissions are difficult to observe. Thus the inversion results are strongly influenced by the high prior estimate from GFEI v1, and are not consistent with the much lower estimate in GFEI

v2. The Qu21 inversion gives lower oil/gas emissions for Russia compared to all bottom-up inventories but we suspect that this reflects their non-optimization of wetlands, which have substantial overlap with oil/gas emissions in Russia. The decreasing trend in Russian gas emissions for 2010-2017 cannot account for differences between GFEI v2 and Lu21.

Lu21 find higher oil/gas emissions for the US and Canada compared to GFEI v1 with high averaging kernel sensitivities for both countries. Many past studies in the US have found an underestimate of oil/gas emissions in the US national inventory (Alvarez et al., 2018; Omara et al., 2018; Cui et al., 2019; Maasakkers et al., 2019, 2021; Rutherford et al., 2021) and similar underestimates have been shown for Canada's national inventory (Johnson et al., 2017; Atherton et al., 2017; Baray et al., 2018, 2021; Chan et al., 2020; Scarpelli et al., 2022; MacKay et al., 2021;

Tyner and Johnson, 2021). These underestimates are not addressed in the more recent versions of the national inventories as used in GFEI v2 (Table 2). The subsector emissions distribution for Canada in GFEI v2 shows a large underestimate of gas transmission emissions compared to Lu21 but better agreement for gas production (Table S1 and S2). Qu21 agree with Lu21 for the US but find much lower emissions for Canada; this again likely reflects errors in satellite observations at high latitudes with spatial overlap between oil/gas and wetland emissions (Scarpelli

et al., 2022).

    Lu21 and Qu21 find large underestimates of oil/gas emissions in the national inventory of Turkmenistan despite its use of oil production emission factors at the higher end of the IPCC range (Ministry of Nature Protection of Turkmenistan, 2015) (Fig. 3). This may reflect anomalous point sources from faulty operations (Varon et al., 2019,

2021; Barré et al. 2021).

    Both inversions also show underestimates of Uzbekistan's gas emissions in all bottom-up inventories (Table 2 and Fig. 6), with the greatest underestimates in the south-central part of the country which contains most of the country's oil/gas production and gas processing infrastructure. The underestimate is larger for GFEI v2 than for GFEI v1

because it uses a more recent UNFCCC report (Uzhydromet, 2021) that estimates 37% lower national oil/gas emissions. The IEA, EDGAR, and TRACE inventories are even lower than GFEI v1 and v2. The higher resolution results of Qu21 feature an offset between the underestimate in the south-central part of the country and a slight overestimate in the western part (Fig. S3). Both versions of GFEI allocate most of Uzbekistan's gas transmission and processing emissions uniformly along pipelines due to a lack of facility data, and this may not properly account for

the density of gas processing sources in central Uzbekistan.

Venezuela's emissions estimated by the inversions are much higher than any of the bottom-up inventories, and this may reflect venting and flaring of associated gas during oil production. Höglund-Isaksson et al. (2017) pointed out that bottom-up inventories often underestimate emissions of associated gas and that practices vary between countries. Despite increased gas collection efforts by the state-owned oil/gas company (República Bolivariana de Venezuela, 2017) and decreasing oil production for 2014-2019 (resulting in the decreasing emissions trend shown in Fig. 4) (EIA, 2021), Lu21 find little change in Venezuela's oil emissions over 2010-2017 and Qu21 estimate similar emissions to Lu21 for 2019.

The inversions find that GFEI overestimates emissions around the Persian Gulf (Fig. S3), including large contributions from Iran and the UAE with high averaging kernel sensitivities in both inversions and smaller contributions from Qatar and Saudi Arabia. The overestimate in the UAE may reflect the nature of its oil production practices where there are a small number of high producing wells. The inversions find an overestimate of Iran's production emissions along the Persian Gulf including offshore emissions but an underestimate of oil/gas production emissions in northern Iran. A similar pattern of underestimated onshore emissions is found in neighboring Iraq, though the inversions still estimate lower emissions than GFEI v2 and this difference cannot be fully attributed to Iraq's increasing emission trend over 2010-2016 (Fig. 4). This may in part be due to low averaging kernel sensitivities preventing divergence from the much lower prior estimate.

The overestimate of oil/gas production emissions in the Persian Gulf reflects a more general pattern of bottom-up inventories overestimating offshore oil/gas production emissions. Qu21 show an overestimate of offshore emissions in GFEI v1 for Côte d'Ivoire while Lu21 results are limited by the coarse resolution. Both inversions find overestimates of emissions in the South China Sea though averaging kernel sensitivities are low. Previous comparisons of top-down and bottom-up estimates found offshore emissions overestimated by bottom-up inventories in the North Sea (Bergamaschi et al., 2010) and Mexico (Shen et al., 2021; Zavala-Araiza et al., 2021). Individual countries may estimate offshore oil/gas production emissions using lower emission factors like those provided by the IPCC (2006), but these emissions are often aggregated in national reports making it difficult for GFEI to differentiate between onshore and offshore wells for spatial allocation of national emissions.

Lu21 find lower emissions than the EDGAR v6, IEA, and Climate TRACE inventories for a number of countries including Nigeria, Kuwait, and Qatar which all have high averaging kernel sensitivities (Fig. 6), though the ability to quantify national estimates for small countries like Kuwait and Qatar is limited by the coarse resolution of the inversion. The upward revision of Nigeria's emissions in its latest UNFCCC report as reflected in GFEI v2 is not supported by the inversions.

**4 Conclusions**

We have updated the Global Fuel Exploitation Inventory (GFEI) for methane emissions from the oil, gas, and coal sectors. GFEI is based on the national inventories reported by individual countries to the United Nations Framework

Convention on Climate Change (UNFCCC), and spatially allocates emissions to infrastructure locations on a 0.1° x 0.1° grid to support inversion of atmospheric methane observations. Our updated GFEI v2 inventory provides annual emissions for 2010-2019 based on recent country reports to the UNFCCC (2021), as well as new oil/gas well data and improved spatial information for Canada, Mexico, and China.

Russia's oil/gas emissions decrease by an order of magnitude in GFEI v2 relative to GFEI v1, while Nigeria's emissions increase by an order of magnitude, reflecting new emission factors used by the national inventories reported to the UNFCCC. Global oil emissions in GFEI v2 decrease from 2010 to 2019 driven in large part by Iran, Libya, and Venezuela. Global gas emissions decrease from 2010 to 2017, mostly driven by Russia, but then increase in 2018 and 2019. Global coal emissions show mixed trends over the time period, mainly driven by China but with sustained decreases in the US and increases in Russia. GFEI v2 global emissions for all sectors are lower than the EDGAR (v6) and IEA inventories though there is considerable variability for individual countries.

We compared GFEI oil/gas emissions to the results of global inversions of satellite (GOSAT) and in situ (GLOBALVIEWplus) observations of atmospheric methane. These inversions find that GFEI oil/gas emissions are underestimated for the US, Venezuela (oil), Uzbekistan (gas), Canada, and Turkmenistan, leading to an underestimate of global gas emissions. Global oil emissions are overestimated in GFEI v1 compared to inversion results, mostly due to Russia. The inversions support the recent downward revision of Russian emissions in its national inventory but not the increase in Nigerian emissions.

There is considerable interest in using satellite observations of atmospheric methane to evaluate and improve the national inventories used for climate policy. The scope of this work was limited by the sparsity of the GOSAT observations and the coarse resolution of the global inversions. New satellite observations from TROPOMI now provide much higher data density though there are still large regional biases in the early-generation methane retrievals (Qu et al., 2021). As the TROPOMI data improve (Lorente et al., 2021), they will prompt finer-resolution inversions to better quantify emissions on national scales and resolve the regional contributions from individual activities. Inverse analyses of TROPOMI data to evaluate the national methane emission inventories reported by individual countries to the UNFCCC, as enabled here by the GFEI spatial gridding, may enable efficient monitoring of national methane emissions from space in pursuit of climate policy.

**Data/Code availability.** GFEI v2 emission grids for 2019 by sector and subsector are available for download from the Harvard Dataverse at https://doi.org/10.7910/DVN/HH4EUM (Scarpelli et al., 2021). The 2010-2018 emission grids are available upon request. The code used for inventory creation is available upon reasonable request.

**Author contributions.** TRS and SG compiled the datasets and created the inventory with assistance from MPS. DJJ and JRW conceived of and provided guidance for the project. XL, ZQ, and YZ provided inversion data and assisted

with data analysis. FR and DG provided TRACE inventory data and feedback on interpretation. TRS prepared the manuscript with contributions from all coauthors.

**Competing interests.** The authors declare that they have no conflict of interest.

**Acknowledgements.** This work was funded by the NASA Carbon Monitoring System, the NASA Advanced Information Systems Technology (AIST) Program, a National Defense Science and Engineering Graduate Fellowship (NDSEG) to TRS, and funding from the NSFC (42007198) for YZ.

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

**Table 1.** Global methane emissions from oil, gas, and coal exploitation (Tg a[-1])

| | GFEI v2 2019 | GFEI v2 2016 | GFEI v1[a] 2016 | Lu et al. (2021)[b] 2010-2017 | Qu et al. (2021)[b] 2019 |
|---|---|---|---|---|---|
| Oil/gas total | 47.6 | 48.9 | 65.9 | 68.5 (0.5) | 54.4 (0.3) |
| Oil | 25.8 | 28.1 | 41.5 | 38.8 (0.5) | 27.4 (0.3) |
| Production[c] | 25.5 | 27.8 | 41.3 | 38.6 (0.5) | 27.2 (0.3) |
| Transport/Refining | 0.3 | 0.3 | 0.2 | 0.2 (0.0) | 0.2 (0.0) |
| Gas | 21.8 | 20.8 | 24.4 | 29.7 (0.4) | 26.9 (0.2) |
| Production[c] | 8.1 | 7.8 | 7.5 | 10.4 (0.5) | 9.1 (0.2) |
| Processing | 2.2 | 2.0 | 2.4 | 3.3 (0.5) | 3.6 (0.2) |
| Transmission[d] | 5.3 | 5.5 | 8.7 | 9.1 (0.1) | 7.7 (0.1) |
| Distribution | 6.2 | 5.5 | 5.7 | 6.9 (0.1) | 6.5 (0.1) |
| Coal | 32.8 | 30.5 | 31.3 | 23.7 (0.5) | 26.0 (0.3) |

[a] Scarpelli et al. (2020a).
[b] Posterior emission estimates from inversions of atmospheric methane observations using GFEI v1 for 2016 as a prior estimate. Averaging kernel sensitivities in parentheses are the diagonal terms of the reduced averaging kernel matrix $A'$ (equation 2). They extend from 0 (no information from the atmospheric methane observations) to 1 (fully informed by the observations).

[c] Including exploration.
[d] Including storage.

**Table 2.** Methane emissions from oil and gas activities by country (Tg a[-1])[a]

| | Oil | | | Gas | | |
|---|---|---|---|---|---|---|
| | GFEI v2 2016 | GFEI v1 2016 | Lu21 2010-2017 | GFEI v2 2016 | GFEI v1 2016 | Lu21 2010-2017 |
| Russia | 1.9 | 20.5 | 12.7 | 2.4 | 4.4 | 3.1 |
| US | 1.8 | 1.8 | 2.3 | 6.0 | 6.6 | 9.8 |
| Venezuela | 3.3 | 3.2 | 7.7 | 0 | 0 | 0 |
| Uzbekistan | < 0.01 | < 0.01 | < 0.01 | 1.7 | 2.7 | 4.3 |
| Canada | 0.75 | 0.88 | 2.0 | 0.79 | 0.78 | 2.0 |
| Turkmenistan | 0.87 | 0.88 | 1.8 | 0.53 | 0.52 | 1.3 |
| Iran | 3.7 | 3.7 | 2.2 | 0.48 | 0.49 | 0.61 |
| Angola | 1.2 | 1.2 | 1.7 | < 0.01 | < 0.01 | < 0.01 |
| Côte d'Ivoire | 0.72 | 0.85 | 0.94 | 0.12 | 0.11 | 0.13 |
| Ukraine | 0.06 | 0.06 | 0.05 | 1.0 | 1.0 | 0.98 |
| Algeria | 0.05 | 0.05 | 0.05 | 1.1 | 1.2 | 0.89 |
| China | 1.0 | 1.0 | 0.64 | 0.12 | 0.11 | 0.08 |
| UAE | 1.3 | 1.3 | 0.68 | 0.07 | 0.07 | 0.03 |
| Nigeria | 2.1 | 0.19 | 0.10 | 1.0 | 0.23 | 0.16 |
| Iraq | 2.8 | 0.04 | 0.05 | 0.02 | 0.01 | 0.06 |

[a] Oil and gas methane emissions by top emitting countries are shown for GFEI v2 for 2016 (this work), GFEI v1 for 2016 (Scarpelli et al., 2020a), and the inversion of Lu et al. (2021; Lu21) for 2010-2017 (8-year average). Emissions by oil/gas subsector are shown in Table S1 and S2. GFEI v2 emissions for 2019 are shown in Fig. 2. US - United States; UAE - United Arab Emirates.

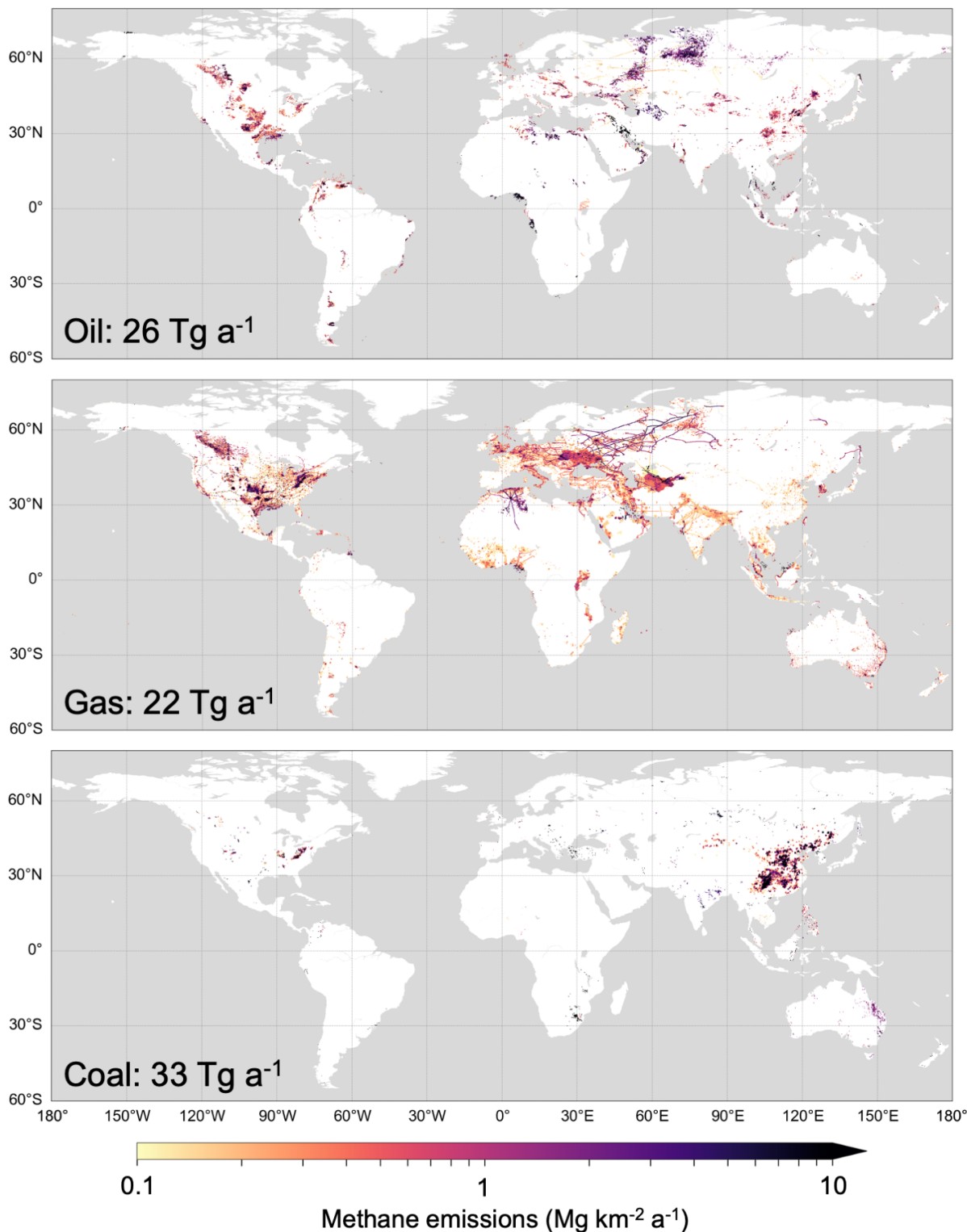

**Figure 1.** Methane emissions from oil, gas, and coal exploitation in GFEI v2 for 2019. Emissions are at 0.1° x 0.1° grid resolution with global emissions inset. Emissions below 0.1 Mg km$^{-2}$ a$^{-1}$ are not shown.


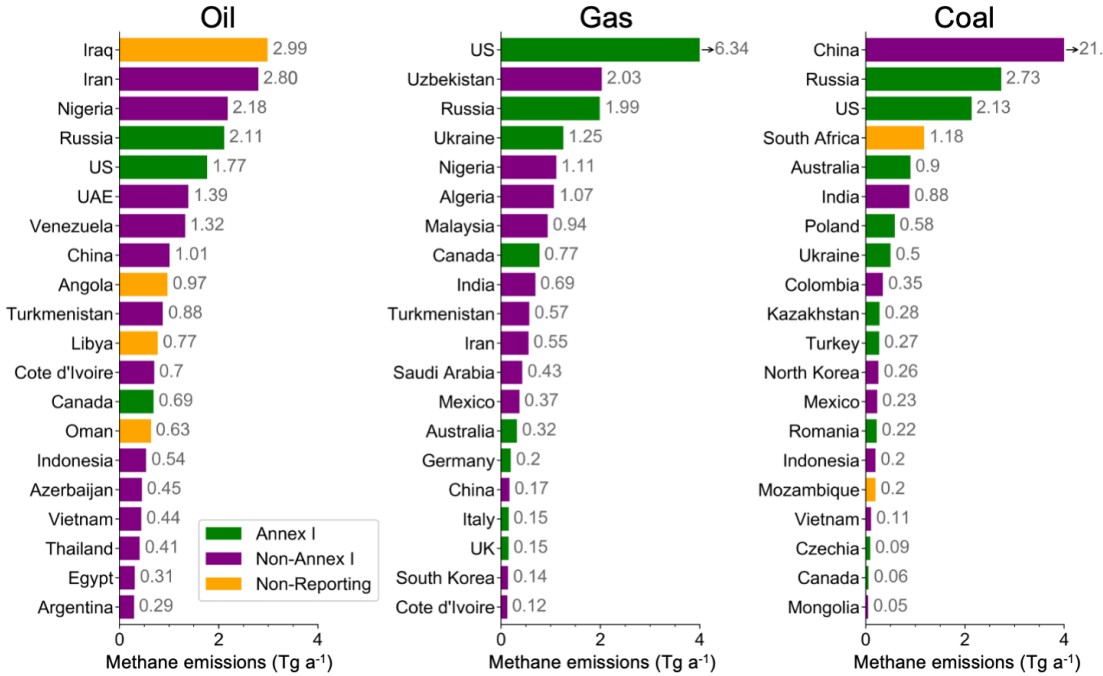

**Figure 2.** Oil, gas, and coal methane emissions by country for 2019 from GFEI v2. Emissions are shown for the top 20 emitting countries. Arrows next to the top bars (highest emitting countries) indicate that emissions are not to scale. US - United States; UAE - United Arab Emirates; UK - United Kingdom.


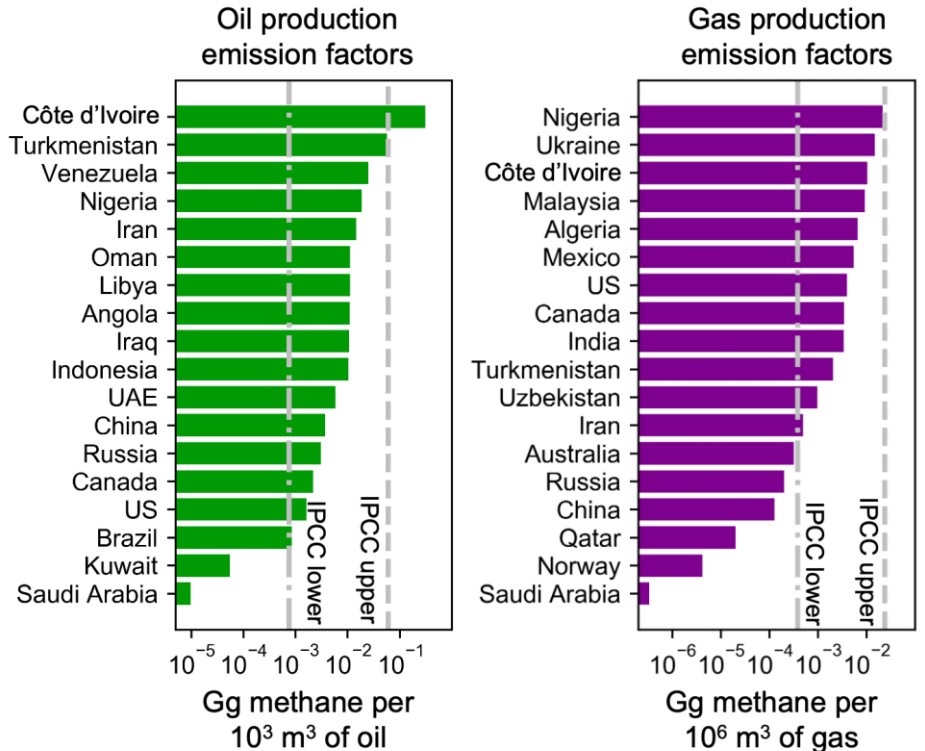

**Figure 3.** Methane emission factors for oil/gas production activities in 2019. Emission factors are shown for the top methane emitting and oil/gas producing countries, and for the IPCC Tier 1 methods (2006). Country emission factors are determined using GFEI v2 oil/gas production emissions and EIA oil/gas production statistics. The IPCC emission factors show the sum of all emission processes (leakage, venting, flaring) with the lower emission factor reflecting the lowest range provided for Developed Countries and the upper emission factor reflecting the highest range provided for Developing Countries and Countries with
Economies in Transition. For oil production, we show emission factors for conventional oil production. US - United States; UAE - United Arab Emirates.

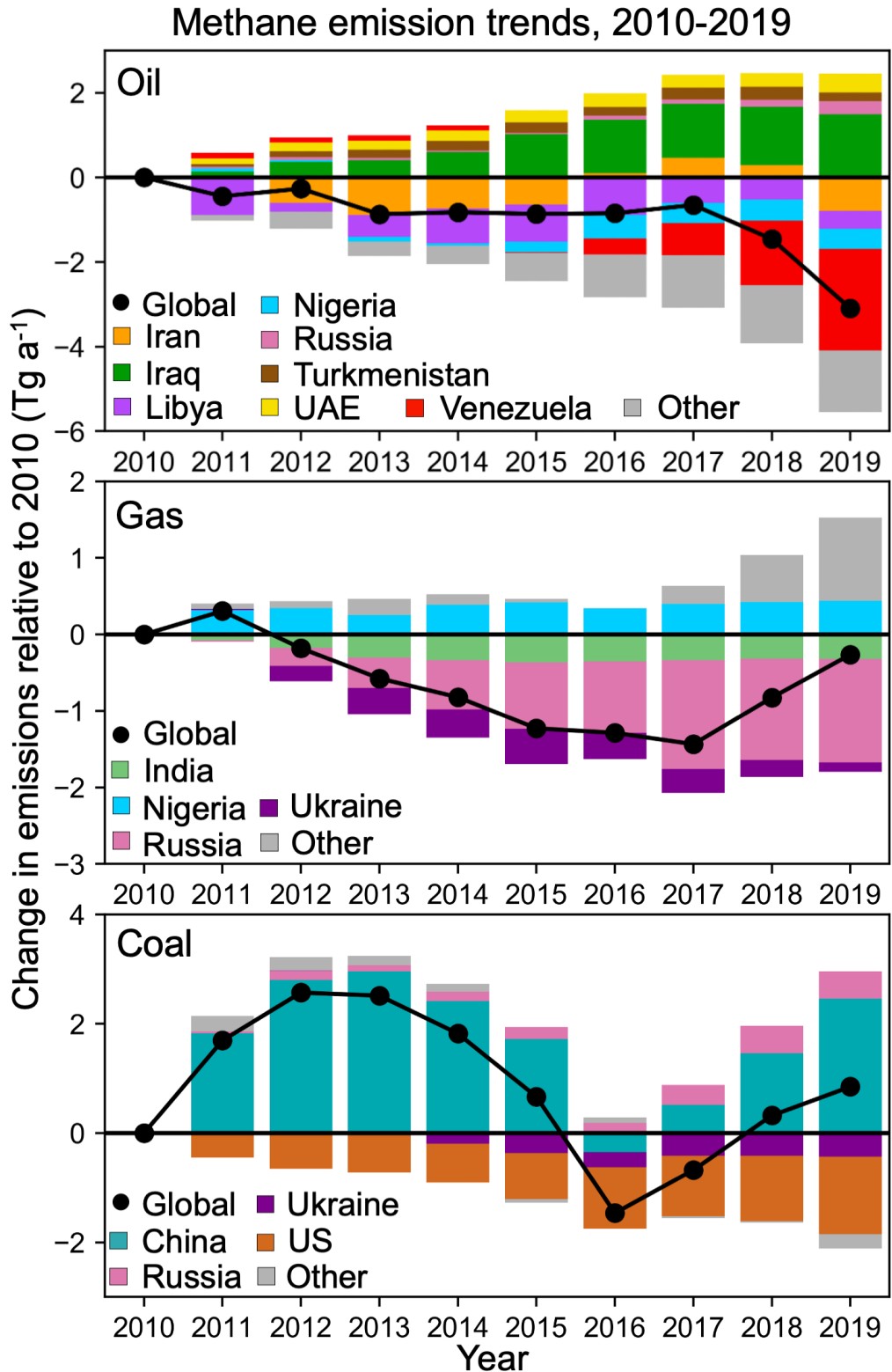

**Figure 4.** Global methane emission trends for the oil, gas, and coal sectors from 2010 to 2019 in GFEI v2, expressed relative to 2010. Trends for individual countries contributing the most to the global trends are also shown. US - United States; UAE - United Arab Emirates.


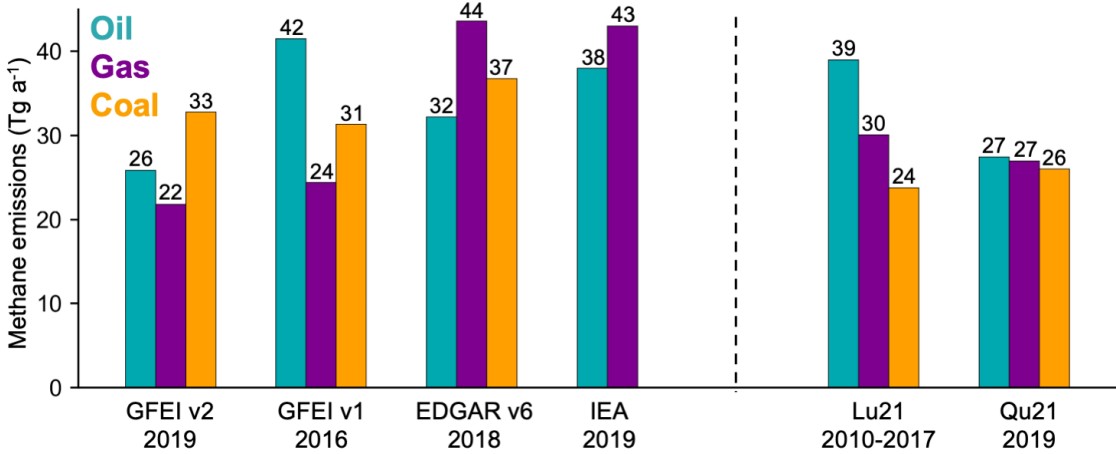

**Figure 5.** Global estimates of oil, gas, and coal methane emissions. The left bars show bottom-up inventories while the right bars show inversion results from Lu et al. (2021; Lu21) and Qu et al. (2021; Qu21).

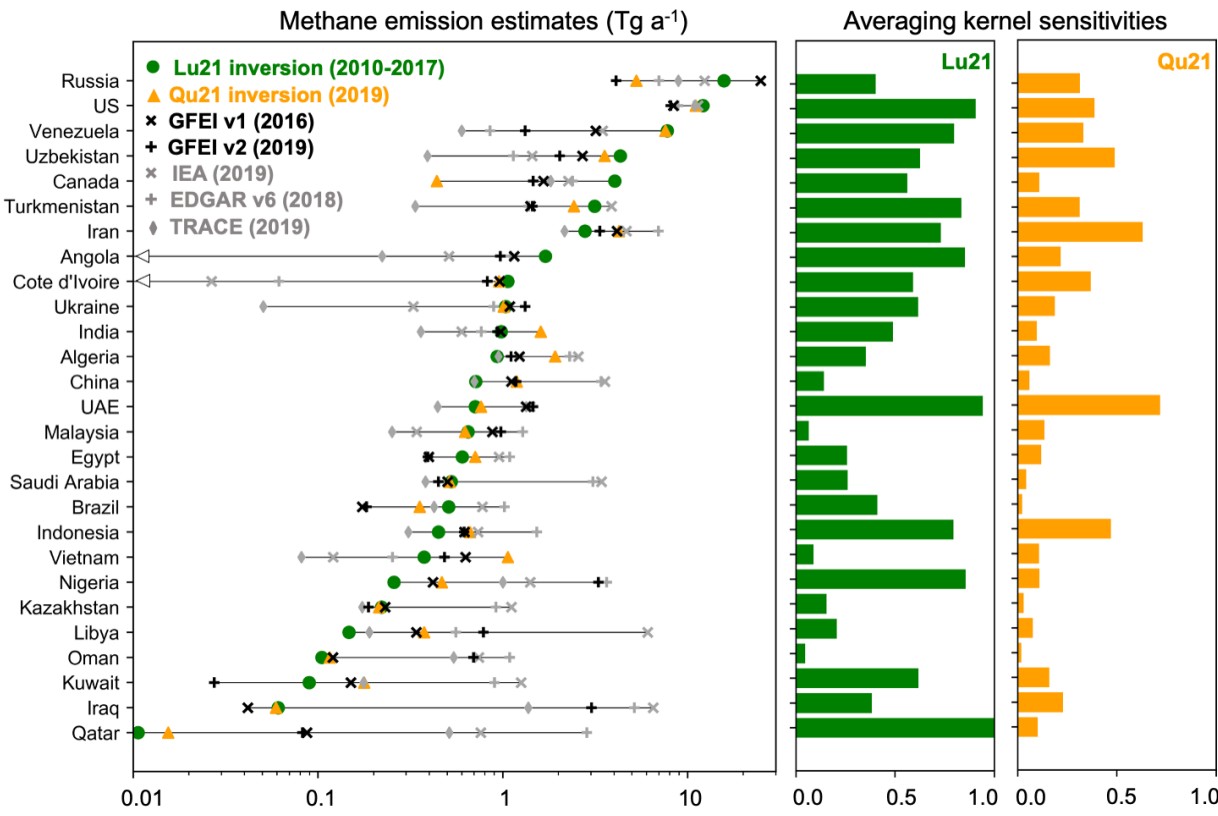

**Figure 6.** Total oil/gas methane emissions in top-emitting countries. The Figure compares bottom-up emissions in the GFEI v1, GFEI v2, EDGAR v5, Climate TRACE, and IEA inventories as well as the inversion results of Lu et al. (2021; Lu21) and Qu et al. (2021; Qu21). Averaging kernel sensitivities for oil/gas emissions in individual countries from the two inversions are also given. The countries shown are those with oil/gas emissions larger than 1 Tg a$^{-1}$ in any of the emission estimates. Horizontal lines extend from the minimum to the maximum emission estimate unless arrow heads designate that emissions in at least one estimate are below 0.01 Tg a$^{-1}$. US - United States; UAE - United Arab Emirates.