# Peer review of "Updated Global Fuel Exploitation Inventory (GFEI) for methane emissions from the oil, gas, and coal sectors: evaluation with inversions of atmospheric methane observations"

_Atmospheric Chemistry and Physics, 2021_

## Author Comment (AC1)

**Response to Referee 1**

We would like to thank Referee 1 for their review. Each referee comment is listed below in italics with the corresponding author response below.

*This manuscript presents updates global inventories for methane emissions from oil, gas, and coal sector (GFEI v2). The inventories are made based on national emissions reported to UNFCCC and shows emissions for years 2010-2019 on 0.1° x 0.1° grid. The manuscript presents changes between GFEI v1 and v2 which come from difference in national emissions reported to UNFCC. The manuscript concentrates mostly on emissions for 2019, but the trend of emissions from oil, gas and coal sector between 2010 and 2019 is also presented. The manuscript also examines results of already published articles of global inversions of atmospheric methane, made based on GFEI v1. The manuscript has detailed methods, results are interesting and presented in a clear way, and is well written. Therefore, I suggest publication after minor revisions.*

Author response: Thank you for the positive review.

*General comments*
*Authors used global inversions based on GOSAT and GLOBALVIEWplus. However, authors do not describe how satellite measurements and in situ observation were made. Please, be aware not all readers know these methods and thus, you should shortly explain how measurements are made using satellite and measurement network.*

Author response: We have added additional explanation of satellite measurements and their importance to the Introduction.

*The general discussion of uncertainties of GFEI is missed, both on global and country scale. Please, specify.*

Author response: We've added a discussion of GFEI v2 uncertainties to Section 2.1.

*The manuscript shows updated version of GFEI inventories. However, the manuscript should be independent from the article describing previous inventories, so readers can easily understand presented work without need to read also the previous version. Please correct the manuscript to make it more independent from the previous article, especially by extending method explanation of construction GFEI v2.*

Author response: Thank you for this suggestions. We have updated the method explanation in Section 2.2 to be more independent of previous work.

*The manuscript should be focused on inventories, including more detailed methods descriptions. However, in present version, the inversion methods are better described, and authors devote more space for inversions, which have already their own publications, than for inventories which should be the core of the manuscript. Please revise the manuscript to keep better proportion between these two parts.*

Author response: We have added further description of the methods for the inventory construction in Section 2.2.

*Compared emissions are calculated for different years and Lu21 inversions are calculated for period 2010-2017. How does it affect comparison and what bias come from comparison of*

*emissions for different years? Also, how does one averaged emission for the period 2010-2017 for Lu21 can affect comparison?*

Author response: We have added discussion of the trends and their impact on our comparisons in paragraph 1 of Section 3.2. In Section 3.2, we also discuss the potential impact of trends on our comparison for individual countries that have large emission trends in GFEI v2, including Venezuela, Iraq, and Russia.

*Specific comments*
*L45: "they may have large uncertainties, particularly for the oil/gas sector" – Where the uncertainties come from and why they are particular for the oil and gas sector?*

Author response: Clarified.

*L73-74:" The atmospheric observations and the transport model are prone to their own errors." Which errors and how they affect calculations? Please specify.*

Author response: We have removed this sentence as it is not the focus of this work.

*L98: How UNFCCC emissions were disaggregated and then allocated to subsector to obtain GFEI v1*

Author response: We have added discussion of our method for this disaggregation in Section 2.2 in the context of GFEI v2, as suggested by the reviewer.

*L120-121: Please add short explanation why Nigeria emissions are higher in the recent national report and how much.*

Author response: We have made it clearer that we explain the reasons for this later in the paper (in our discussion of Figure 3).

*L123: "we start from the same spatial infrastructure" – what exactly?*

Author response: Clarified.

*L177: Please describe what is emission factor*

Author response: Clarified.

*L226: Which global inversions and for which year?*

Author response: This is discussed in great detail in the methods (Section 3.1).

*L302: Which inversions?*

Author response: Clarified.

*L303: Please describe more dependance of the inversion results on priori estimate, e.g., what affect this dependency, what is a role of wetlands?*

Author response: Addition description provided in the text.

*Figure 6 and everywhere else: Lu21 made inversion calculation for period 201-2017, which value was taken to compare with others? The average for a whole period or something else?*

Author response: Clarified in the text.

*L321: why GFEI v1 and v2 underestimates emissions? Is it possible that rather inversion overestimate emissions?*

Author response: The bottom-up inventories have large uncertainties (as we mention in the Introduction), so through the inversion we gain additional information from the observational data (and the associated errors) and that is why we treat the results as corrections to the bottom-up inventory. We have clarified this at the beginning of Section 3. It is possible that the higher emission fluxes from the inversion are related to a different source of emissions, as we briefly in the context of wetlands.

*L364: Should be added: than used for GFEI v1?*

Author response: Clarified.

*L366: Both GFEI v1 and v2?*

Author response: Clarified.

**Response to Referee 2**

We would like to thank Referee 2 for their review. Each referee comment is listed below in italics with the corresponding author response below.

*The manuscript presents an update on the Global Fuel Exploitation Inventory (GFEI) that estimates methane emissions from oil, gas and coal sectors and maps it on a 0.1 \* 0.1 degree grid. The update includes annual trends in emissions from 2010-2019 that is very useful in understanding the changes in oil and gas activities and abatement measures from each country over time. The two versions of GFEI are compared to other bottom up emission inventories such as EDGAR and IEA as well as two top down emission estimates based on satellite observation inversions previously done by Lu et al. (2021) and Qu et al. (2021). Incorporating satellite inversion estimates to assess the bottom-up estimates submitted to the UNFCCC by each country is a useful way to independently validate these reports.*

Author response: Thank you for the positive comment.

*Here are some general comments that I think would help improve the manuscript:*
1. *It is mentioned that bottom up emission inventories have large uncertainties but top down inversions are also prone to their own errors. However you haven't mentioned the order of magnitude of the uncertainty in each case. I suggest you add uncertainty values to table 1 and table 2. If the uncertainty in top down inventories are lower than the bottom up in that case we can state the bottom up estimates are underestimated or overestimated. Otherwise using the term bias is more appropriate as we don't know which one is closer to the truth.*

Author response: Thank you for this suggestion. We have added discussion of inventory uncertainties in Section 2.1. We discuss the ability of the inversions to reduce our emission estimate uncertainty through our discussion of the averaging kernel sensitivities.

2. *Figure 5. Could you use the trends in Figure 4 to project emissions from all the inventories to a reference year? That way the comparisons would become more meaningful.*

Author response: We have added discussion of the impact of trends to the text. We do not expect trends to have a large impact on the comparison as the trends (on the order of 2 Tg globally for oil/gas) are much smaller than inventory differences.

3. *Line 307. You mentioned earlier that observations from latitudes higher than 60 are excluded from both inversions. Given there are wetlands in these regions, wouldn't both models heavily rely on wetland priors in Canada and Russia?*

Author response: Yes, the inversions rely heavily on the prior estimate higher than 60 degrees N latitude, though neighboring observations can provide some information. Russia and Canada also have wetlands below this latitude that overlap with oil/gas, so we do get information from the observations in Russia and Canada which drives the statement referred to.

*Minor corrections and comments:*
- *Line 33. ...Nigerian emissions are too high. This sentence is a bit vague. Nigerian emissions are too high in the inversion results or UNFCCC report?*

Author response: We have clarified this in the text.

- *Line 108. Do you mean total emissions used for Canada, Mexico and US are from UNFCCC but the distribution is scaled according to Sheng et al and Maasakkers et al? Please clarify.*

Author response: We have changed this sentence to better clarify our meaning.

- *Line 121. Can you bring the updated emission value from Nigeria? (instead of stating much higher)*

Author response: We do not wish to state national emissions in the Methods section so we have adjusted the sentence to simply point out that Nigeria is now above our 1 Tg $a^{-1}$ limit.

- *Line 128. This sentence is a big vague. Are you referring to improved emissions in distribution of "coal" emissions in Canada and Mexico? Can you explain in what way they are improved?*

Author response: We have clarified this in the text.

- *Line 140. ..oil emissions more distributed among the top emitting countries. Do you mean more evenly distributed?*

Author response: Yes, clarified.

- *Line 301. You are introducing a new inventory here. Can you add some information about the Climate TRACE inventory in the introduction?*

Author response: We have added a more detailed description of the Climate TRACE inventory in Section 3.2.